# Eicosanoids in Skin Wound Healing

**DOI:** 10.3390/ijms21228435

**Published:** 2020-11-10

**Authors:** Ken Yasukawa, Toshiaki Okuno, Takehiko Yokomizo

**Affiliations:** 1Department of Biochemistry, Juntendo University Graduate School of Medicine, Tokyo 113-8421, Japan; k-yasukawa@juntendo.ac.jp (K.Y.); yokomizo-tky@umin.ac.jp (T.Y.); 2Drug Discovery Research Department, Sato Pharmaceutical Co., Ltd., Tokyo 140-0011, Japan

**Keywords:** eicosanoids, wound healing, lipid mediators, inflammation, specialized pro-resolving mediators

## Abstract

Wound healing is an important process in the human body to protect against external threats. A dysregulation at any stage of the wound healing process may result in the development of various intractable ulcers or excessive scar formation. Numerous factors such as growth factors, cytokines, and chemokines are involved in this process and play vital roles in tissue repair. Moreover, recent studies have demonstrated that lipid mediators derived from membrane fatty acids are also involved in the process of wound healing. Among these lipid mediators, we focus on eicosanoids such as prostaglandins, thromboxane, leukotrienes, and specialized pro-resolving mediators, which are produced during wound healing processes and play versatile roles in the process. This review article highlights the roles of eicosanoids on skin wound healing, especially focusing on the biosynthetic pathways and biological functions, i.e., inflammation, proliferation, migration, angiogenesis, remodeling, and scarring.

## 1. Introduction

Skin is the largest organ in the entire body, which acts as a physical, chemical, biological, radiological, and thermal barrier system and prevents dehydration from the body [1,2]. Wound healing is an important but complicated process in the body that helps protect against external threats [3]. This process involves three consecutive and overlapping steps, including hemostasis/inflammatory, proliferative, and remodeling phases [4]. An impairment of this process leads to the formation of various intractable ulcers such as diabetic ulcers and pressure ulcers [5]. Conversely, excessive wound healing results in the formation of hypertrophic scars or keloids [6,7,8]. Complete integral wound healing requires the cooperative interactions of numerous factors and multiple cell types. Studies have documented that growth factors, cytokines, and chemokines play vital roles in wound healing processes [9,10,11]. Moreover, recent studies have demonstrated that various lipid mediators such as lysophosphatidic acid, sphingosine-1-phosphate, and eicosanoids are also key players in this process [12,13,14,15,16].

Eicosanoids are one of the most important lipid mediators that are rapidly produced upon cell activation by enzymatic conversion of polyunsaturated fatty acids (PUFAs) with 20 carbon atoms, which are derived from membrane phospholipids by the hydrolytic activity of phospholipase A_2_ (PLA_2_), particularly cytosolic PLA_2_α [17,18]. Among eicosanoids, metabolites derived from omega-6 PUFA arachidonic acid (AA) have been well described in the context of the homeostasis and immune system [19,20]. AA can be metabolized into prostaglandins (PGs) and thromboxane A_2_ (TxA_2_) by the cyclooxygenase (COX) pathway [21,22]; leukotrienes (LTs), hydroxyeicosatetraenoic acids (HETEs), and lipoxins (LXs) by the lipoxygenase (LOX) pathway [23,24]; or epoxyeicosatrienoic acids (EETs) by the cytochrome P450 (CYP) pathway [25,26]. These metabolites are produced in a time-dependent manner after injury and can act as either a positive or negative regulator of wound healing. Furthermore, recent studies have reported on biological activities that promote wound healing by specialized pro-resolving mediators (SPMs), such as resolvin series derived from the omega-3 PUFA eicosapentaenoic acid (EPA) or docosahexaenoic acid (DHA) [27,28].

This review is intended to summarize the biosynthetic pathways and functions of eicosanoids and SPMs in skin wound healing. We also discuss the therapeutic potential of targeting eicosanoid and SPM signaling as a treatment approach for wound healing. A comprehensive understanding of complex eicosanoid interactions would help provide novel insights into the development of novel and precise therapeutic approaches for patients with impaired wound healing.

## 2. Biosynthetic Pathway of Eicosanoids and SPMs

PLA_2_s hydrolyze the *sn-*2 position of phospholipids, which are the major components of cell membranes, and liberate free fatty acids and lysophospholipids [29]. In response to various stimuli, including tissue damage, PUFAs, including AA, EPA, and DHA, are released from membrane phospholipids and can be metabolized into various bioactive lipid mediators. Several enzyme families such as COXs, LOXs, and CYPs are implied in the conversion of PUFAs (Figure 1).

Prostanoids are formed by the majority of cells in our bodies through the action of COX enzymes and participate in a variety of physiological and pathological processes [30]. The family of bioactive prostanoids includes PGs (PGD_2_, PGE_2_, PGF_2_α), prostacyclin (PGI_2_), and TxA_2_ [31]. COX-1 is constitutively expressed in almost all cell types, whereas COX-2 is inducible by several stimuli, including pro-inflammatory and mitogenic signals, thus implying its role in inflammation and cell proliferation [32]. Through the action of COX enzymes, AA is oxygenated into a hydroperoxy endoperoxide PGG_2_ and subsequently reduced to a hydroxy endoperoxide PGH_2_. The synthetic activity of COXs can be inhibited by nonsteroidal anti-inflammatory drugs (NSAIDs), especially celecoxib, a COX-2-selective inhibitor with reduced side effects [33]. In the skin, COX-2 is induced by various stimuli such as injury, mechanical scratching, ultraviolet B irradiation, and cancer [34,35,36,37]. AA is converted into PGH_2_ by COXs and then into prostanoids by the activity of a range of terminal synthases [38].

The products generated by the LOX pathway are important regulators of innate immunity and inflammation, which can promote both pro-inflammatory and anti-inflammatory responses [39,40]. Mammals have several types of LOX isoforms [41]. AA can be oxygenated by 5-LOX to yield 5-hydroperoxyeicosatetraenoic acid (5-HpETE) and then converted into an unstable intermediate, leukotriene A_4_ (LTA_4_). LTs consist of two groups, LTB_4_ and peptide-conjugated cysteinyl LTs (CysLTs; LTC_4_, LTD_4_, and LTE_4_). LTB_4_ and LTC_4_ are produced from LTA_4_ by the action of LTA_4_ hydrolase (LTA_4_H) and LTC_4_ synthase (LTC_4_S), respectively, and LTC_4_ can be further metabolized into LTD_4_ and LTE_4_ [42]. Alternatively, HpETEs generated by LOXs can be reduced by peroxidases to the monohydroxy fatty acids HETEs. The frequent sites of oxidation of AA are 5-, 8-, 12-, and 15-carbon positions, and these reactions are catalyzed by 5-, 8-, 12-, and 15-LOX enzymes, respectively. The oxidation at another position can be catalyzed by COX and CYP enzymes [24]. Moreover, the sequential reaction of 5-LOX and 12- or 15-LOX generates LXs (LXA_4_ and LXB_4_) from AA through interactions with leukocytes and other cells such as platelets and epithelial cells. LXs are categorized as a member of SPMs and act to resolve inflammatory responses [43,44].

EETs are epoxide derivatives of AA formed through the action of CYP epoxygenases, which are abundant in endothelial cells. There are four isomers of EET, 5,6-, 8,9-, 11,12-, and 14,15-EET, that function as autocrine and paracrine mediators and are implicated in vascular relaxation, anti-inflammatory effects, and angiogenesis. Soluble epoxide hydrolase (sEH), which metabolizes EETs into inactive dihydroxyeicosatrienoic acids (DHET), attenuates several functional effects of EETs [45].

In addition to AA, LOXs and CYPs can metabolize the omega-3 PUFA EPA and DHA and can produce several classes of SPMs (Figure 2). E-series of resolvins (RvE1, RvE2, and RvE3) are formed from EPA through the combination of LOX and CYP enzymes, whereas D-series of resolvins (RvD1, RvD2, RvD3, RvD4, RvD5, and RvD6) are formed from DHA through the combination of LOX enzymes. Moreover, DHA can be metabolized into another class of SPMs such as protectins (PD1 and PDX) and maresins (MaR1 and MaR2) [28,46].

Altogether, in the context of wound healing, a wide variety of eicosanoids and SPMs are produced by the wound stimuli and play multiple roles in the wound healing process (Figure 3). The detailed functions of eicosanoids and SPMs are described in the following sections.

## 3. Functions of Eicosanoids and SPMs in Skin Wound Healing

### 3.1. COX Metabolites

#### 3.1.1. TxA_2_

The cellular source of each eicosanoid is dependent on organs and cell types [47]. In the context of wound healing, TxA_2_ is abundantly produced by activated platelets through the action of TxA_2_ synthase (TxAS) immediately after the injury and contributes to platelet activation and irreversible platelet aggregation for hemostasis [48,49]. In fact, the lack of G protein-coupled thromboxane-prostanoid (TP) receptor in mice resulted in prolonged bleeding time and TP agonist caused no detectable aggregation of these mice platelets [50]. Administration of TP antagonist was found to significantly inhibit platelet aggregation in humans [51]. Platelet aggregation is an essential system to stop bleeding to protect our body against external threats and is involved in efficient wound healing [52,53].

In addition, TxA_2_ produced by activated platelets was found to induce the synthesis of the pro-inflammatory cytokine interleukin (IL)-6 and PGE_2_ and suppress the expression of the anti-inflammatory macrophage marker CD206 in macrophages through the activation of the TP receptor in a cutaneous inflammation mouse model [54]. Moreover, studies have demonstrated that TxA_2_ can be synthesized in microvascular endothelial cells under COX-2-induced condition and promotes endothelial migration and angiogenesis [55,56].

Altogether, the TxA_2_/TP axis might be involved in wound healing as a platelet aggregant in the hemostasis/inflammatory phase as well as a mediator of inflammation and tissue regeneration in subsequent phases.

#### 3.1.2. 12(*S*)-Hydroxyheptadeca-5*Z*,8*E*,10*E*-Trienoic Acid (12-HHT)

TxAS catalyzes the conversion of PGH_2_ into 12-HHT and malondialdehyde in an equimolar ratio into TxA_2_. Although 12-HHT had long been considered as merely a by-product of TxA_2_ synthesis, recent studies have demonstrated that it is a natural endogenous ligand for the low-affinity LTB_4_ receptor BLT2 [57]. BLT2 is abundantly expressed in epidermal keratinocytes, and the 12-HHT/BLT2 axis promotes keratinocyte migration through the production of tumor necrosis factor α (TNFα) and matrix metalloproteinases (MMPs) and thereby accelerates skin wound closure in mice [58].

In addition, it has been reported that BLT2 agonist directly promotes keratinocyte migration and indirectly enhances fibroblast proliferation by increasing the keratinocyte production of transforming growth factor-β1 (TGF-β1) and basic fibroblast growth factor (bFGF) and thus accelerated wound closure in a rat model of streptozotocin-induced diabetes [59]. Importantly, the delay in wound healing caused by the administration of NSAIDs represented by aspirin was abrogated in BLT2-deficient mice.

These results suggest a novel mechanism underlying the aspirin-dependent delay in wound healing and provide a promising novel therapeutic potential of BLT2 agonist for the treatment of intractable ulcers [58,60,61].

#### 3.1.3. PGE_2_

As mentioned earlier, COX-2 is an inducible enzyme and rapidly induced by the stimulation of injury, leading to the enhanced production of PGs, including PGE_2_. It has been reported that cytosolic PGE_2_ synthase (cPGES) can be coupled with COX-1 and participates in the maintenance of tissue homeostasis and immediate PGE_2_ synthesis [62]. In contrast, membrane-associated PGE_2_ synthases (mPGESs) are functionally coupled with COX-2 and essential components for delayed PGE_2_ biosynthesis, which may be associated with inflammation [63]. PGE_2_ is the most abundant PG in various tissues and plays versatile roles in physiological and pathological actions by activating four E-prostanoid (EP) receptors (EP1-4), which are categorized as G protein-coupled receptors (GPCRs) [64].

In the early phase of wound healing, immigrating macrophages and stromal cells produce PGE_2_, which induces angiogenesis and proliferation of human fibroblasts through the induction of bFGF in fibroblasts, thereby accelerating wound repair [65,66]. In fact, in diabetic *ob/ob* mice, the concentration of PGE_2_ was reduced in the cutaneous wound [67]. In association with that study, prostaglandin transporter (PGT), which mediates PG catabolism and degradation in the cytosol, is upregulated by hyperglycemia, resulting in diminished PGE_2_ signaling [68].

On the other hand, PGE_2_ and its metabolite, 13,14-dihydro-15-keto-PGE_2_, which are rich in wound fluid, induce the expression of oncostatin M (OSM) in wound-site macrophages through the activation of EP4. OSM is structurally and functionally related to the IL-6 family and acts as an anti-inflammatory cytokine by attenuating the expression of TNFα and IL-1β, thereby improving wound healing [69]. Moreover, studies have reported that PGE_2_ affects the proliferation and differentiation of keratinocytes through EP2 and EP4, which increase intracellular cAMP concentration [70,71,72,73,74].

In the late phase of wound healing, the transition of the pro-inflammatory M1 phenotype macrophage to M2 regenerating phenotype is required to achieve normal wound healing, and impaired switching of M1 to M2 phenotype could lead to the formation of a nonhealing wound [75]. PGE_2_ promotes M2 macrophage polarization through the cyclic AMP-responsive element binding (CREB)-mediated induction of Krupple-like factor 4 (KLF4) [76]. In fact, the topical application of PGE_2_ containing chitosan hydrogel was found to accelerate wound closure by ameliorating inflammation by promoting the M2 phenotypic transformation of macrophages [77].

Moreover, the PGE_2_/EP2 axis regulates the balance of MMPs and the tissue inhibitor of MMP (TIMP), resulting in the inhibition of the TGF-β1-induced collagen synthesis in dermal fibroblasts, thereby reducing hypertrophic scar formation [78].

#### 3.1.4. PGD_2_

The synthesis of PGD_2_ from PGH_2_ is manipulated by two PGD synthase (PGDS) subtypes, hematopoietic-type (H-PGDS) and lipocalin-type (L-PGDS), which are evolutionarily different from each other [79]. Although the pro-inflammatory and anti-inflammatory functions of PGD_2_ are well known, its direct function in wound healing remains incompletely understood [80]. In the skin, PGD_2_ mitigated the pruritic behavior in an atopic dermatitis mouse model and accelerated the recovery of cutaneous barrier function after mechanical scratching through the activation of the D-prostanoid (DP) receptor DP1 [81,82,83].

Studies have well established that PGD_2_ and its J-ring-type metabolites, especially 15-deoxy-Δ^12,14^-prostaglandin J_2_ (15d-PGJ_2_), act as a ligand for peroxisome proliferator-activated receptor γ (PPARγ) [84,85]. The activation of PPARγ in monocytes induced the differentiation into anti-inflammatory M2 macrophages, whereas the genetic deletion of PPARγ in macrophages resulted in impaired wound healing, suggesting that PGD_2_ and its metabolites participate in the resolution of inflammation and promotion of wound closure [86,87]. In fact, it has been demonstrated that the topical application of 15d-PGJ_2_ to the cutaneous wound of diabetic *db/db* mice enhanced the activity of PPARγ in wound-site macrophages and accelerated wound closure with reduced inflammation [88].

In contrast, the PGD_2_/DP1 axis diminished the migration of fibroblasts [89]. Although fibroblast migration is essential for tissue repair, excessive accumulation of fibroblasts can cause aberrant tissue reconstitution such as fibrosis or keloids, thus implying the modulating function of PGD_2_ in tissue regeneration.

Moreover, the PGD_2_/DP1 axis is known to exert suppressive functions in VEGF- or IL-1β-induced vascular leakage and angiogenesis [90]. Consistent with the suppressive effects, L-PGDS-derived PGD_2_ was found to inhibit hair follicle neogenesis through DP2 receptor (also known as a chemoattractant receptor homologous molecule expressed on type 2 T-helper cells; CRTH2 or GPR44) signaling during wound repair [91]. In fact, both L-PGDS gene expression and PGD_2_ level are significantly elevated in the bald scalp compared to those in the haired scalp in humans, and the topical application of PGD_2_ inhibited hair growth in both mice and humans through DP2 signaling [92].

#### 3.1.5. PGF_2_α

Since several decades ago, the PGF_2_α/F-prostanoid receptor (FP) signal has been known to induce labor and control intraocular pressure [93,94]. Although several studies have demonstrated that PGF_2_α is produced in the sites of injury, atopic dermatitis, and psoriasis, the function and significance of FP in the skin are less understood [95,96,97]. Recent research has reported the hypertrichotic effect of FP agonist, thereby implying the involvement of PGF_2_α in tissue remodeling; however, the molecular mechanisms underlying the hair growth remain unknown [98].

#### 3.1.6. PGI_2_

PGI_2_ (also known as prostacyclin) is primarily produced in endothelial cells by the action of prostacyclin synthase (PGIS) from PGH_2_ and is rapidly converted by nonenzymatic processes into an inactive metabolite, 6-keto PGF_1_α [99]. It is well known that PGI_2_ acts as a vasodilator, maintaining the appropriate blood flow to peripheral tissues [100]. Although hemostasis is an indispensable mechanism required for regular wound healing as described earlier, it is important to remove the clot for the subsequent process. In fact, stimulation of I-prostanoid receptor (IP) promotes fibrinolysis by enhancing the production of urokinase-type plasminogen activator (uPA) in fibroblasts and accelerating fibroblast migration [101].

On the other hand, interestingly, the PGI_2_ analogs iloprost and carbaprostacyclin, which activate both IP and PPARδ, are known to have angiogenic function through the induction of vascular endothelial growth factor (VEGF), whereas the IP-specific agonist cicaprost does not possess such function [102,103].

### 3.2. LOX Metabolites

#### 3.2.1. Leukotrienes (LTB_4_ and CysLTs; LTC_4_, LTD_4_, and LTE_4_)

LTB_4_ binds to its receptor BLT1, and the most important function of the LTB_4_/BLT1 axis is its potent chemotactic effect for several immune cells [40,104]. In the initial inflammatory phase of wound healing, inflammatory cells, especially neutrophils, are recruited to numerous chemotactic factors generated from injured tissues to clean up the injured sites and initiate subsequent inflammatory reactions. However, although the inflammatory process is an important component of the wound repair process in response to tissue damage, excessive production of LTB_4_ has been found to cause uncontrolled neutrophil chemotaxis and insufficient bacterial clearance in the skin of diabetic mice [105]. BLT2 was initially identified as a low-affinity receptor for LTB_4_, but it is now considered as the receptor for 12-HHT as mentioned earlier [57,106].

CysLTs are produced primarily by activated eosinophils, basophils, mast cells, and macrophages and are involved in bronchoconstriction and allergic response. CysLT receptor (CysLT1 and CysLT2) antagonists are reported to be clinically efficacious in patients with asthma or allergic rhinitis [107]. Moreover, recent studies found GPR99 as a potential novel receptor for LTE_4_, which is the most stable and abundant CysLT at the site of inflammation, and the intradermal injection of LTE_4_ induced vascular permeability in wild-type mice ear but not in GPR99-deficient mice ear [108,109].

Several studies have demonstrated that the 5-LOX/LTs axis is involved in impaired cutaneous wound healing. Genetic deficiency or pharmacological inhibition of 5-LOX in mice resulted in decreased reactive oxygen species (ROS) formation in fibroblasts by upregulating the expression of heme oxygenase-1 (HO-1), thereby accelerating the process of wound repair [110]. Furthermore, BLT1 antagonists or CysLT receptor antagonists, as well as 5-LOX deficiency, was found to improve wound closure by modulating the inflammatory response [111]. In addition, increased systemic levels of LTB_4_ and pro-inflammatory cytokines were observed in streptozotocin-induced type 1 diabetic mice with delayed wound healing compared to those in healthy mice. Genetic 5-LOX deletion in diabetic mice resulted in improved wound healing with increased frequency of alternative M2 macrophage population [112].

On the other hand, it has been reported that CysLTs directly promoted collagen production, and IL-6 and granulocyte macrophage colony-stimulating factor (GM-CSF) are released from fibroblasts through the activation of CysLT2 but not CysLT1, thereby inducing keratinocyte proliferation in ovalbumin-sensitized skin [113].

Altogether, these studies suggest that the inhibition of 5-LOX activity or the selective antagonism of BLT1 or CysLTs is beneficial for mitigating inflammation and could be a therapeutic alternative for patients with unresolved cutaneous healing such as a diabetic ulcer.

#### 3.2.2. HETEs

The production of HETEs from AA is manipulated by the activity of various LOXs. The major HETEs identified in the human skin are 12- and 15-LOX products [114]. Although the presence of wide variations of HETE isoforms in the skin is known, the physiological roles of HETEs in wound repair largely remain unknown. It has been reported that 12-HETE treatment enhanced keratinocyte proliferation and chemotaxis to 12-HETE, implicating its participation in wound healing [115,116]. BLT2 is known to be activated by high 12-HETE concentration [117]; however, its role in the skin function of 12-HETE is unknown.

### 3.3. CYP Metabolites (EETs)

CYP epoxygenases (CYP2C8 and CYP2J2 in humans) are primarily expressed in endothelial cells and can catalyze the epoxidation of AA to produce EETs. It has been reported that EETs are bioactive lipid mediators that possess vasodilative, anti-inflammatory, and angiogenic effects [45]. In addition, recent studies have demonstrated their direct function in the promotion of skin wound healing.

The forced expression of human CYP2C8 or CYP2J2 in mice endothelial cells (Tie2-promoter driven) and the deficiency of sEH resulted in accelerated cutaneous wound closure [118,119]. In the same manner, the systemic administration of 11,12-EET, 14,15-EET, and sEH inhibitor displayed similar effects, including increased vascularization, compared to those in control mice as evaluated by histological analyses [118]. Topical application of 11,12-EET or 14,15-EET in mice ears was found to ameliorate ischemic wound healing with a significant elevation of the expression of VEGF, TGF-β1, and stromal cell-derived factor 1α (SDF-1α) [120]. The expression of CYP2C65 and CYP2J6 was significantly reduced in the granulation tissues of diabetic *ob/ob* mice, and the injection of 11,12-EET resulted in mitigated inflammatory reactions and increased collagen deposition, leading to accelerated wound healing [121]. Moreover, the administration of sEH inhibitor or a combination of sEH inhibitor and EETs was found to improve the engraftment of transplanted skin graft with increased vascularization [122].

These studies suggest the therapeutic potential of the treatment of EETs and/or sEH inhibitor with the functions of neovascularization and anti-inflammatory effect in various pathologies of wound healing.

### 3.4. SPMs (Lipoxins, Resolvins, Protectins, and Maresins)

SPMs are also produced through the metabolism of PUFAs and implicated in the orchestration of resolution of inflammation. Recent studies suggest that numerous chronic inflammatory pathologies are caused by the incomplete resolution of inflammation, and stimulating the resolution pathways with SPMs could be beneficial in multiple disease states, including skin injury [123,124].

Topical application of LXA_4_ or microparticle-encapsulated LXA_4_ to the dorsal wound of a rat model promoted wound closure with attenuated inflammatory responses and enhanced angiogenesis and collagen accumulation through its receptor ALX [125]. Furthermore, LXA_4_ directly modulated fibroblast proliferation and migration, thus implying the promotion of wound healing without scarring [126].

It has been reported that RvE1 binds to BLT1 as a partial agonist and attenuates LTB_4_/BLT1 signals to modulate leukocyte infiltration as well as stimulating the RvE1 receptor ChemR23 to regulate migration and cytokine production of macrophages and dendritic cells [127,128]. RvE1, RvD1, and RvD2 accelerated dorsal wound closure with a more mature collagen organization upon topical application [129]. Moreover, treatment with RvD1, RvD2, or RvD4 promoted re-epithelialization during skin injury and thus accelerated the process of wound healing. The enhancement of re-epithelialization by RvD1 or RvD2 treatment was abolished by the genetic deficiency of RvD1 receptor (ALX/FPR2) or RvD2 receptor (DRV2/GPR18), which are expressed on keratinocytes, respectively. It has been reported that the PI3K-AKT-mTOR-S6 pathway is involved in the downstream of RvD2 signal [27]. In addition, RvD2 was found to inhibit TGF-β-induced fibroblast proliferation and migration in an in vitro scratch model, thus suggesting its modulating function in the formation of fibrosis [126].

Although PD1 production in diabetic mice wound was decreased, PD1 treatment enhanced the process of wound healing by promoting nerve fiber growth, re-epithelialization, collagen deposition, and reduced inflammation [130].

Injection of MaR1 was found to exhibit a protective function in mice skin against UVB irradiation-induced edema, neutrophil recruitment, collagen degradation, and pro-inflammatory cytokine induction [131]. Another study reported that topical application of MaR1 onto tooth extraction sockets accelerated wound closure by promoting re-epithelialization and inducing an increased ratio of CD206-positive M2-like macrophages [132].

Altogether, recent studies have demonstrated the beneficial effects of the treatment of SPMs in the process of skin wound healing, particularly in conditions that impair the resolution of inflammation, such as diabetes. Development of treatment strategies for wound healing based on SPM pathways may offer distinct advantages over traditional anti-inflammatory therapies that disturb tissue repair.

## 4. Summary and Perspectives

Wound healing is a complex process involving multiple factors such as growth factors, cytokines, chemokines, and lipid mediators at each stage of healing. Despite the increase in the number of patients with impaired wound healing, there are limited mechanism-based therapeutic alternatives for the treatment of intractable wounds. For instance, recombinant human bFGF (Fiblast Spray), which is indicated for decubitus and skin ulcers, including burns and leg ulcers, is available [133]. Therefore, it is necessary to develop treatment options from different perspectives.

In this review, we have summarized the biosynthetic process and functions of lipid mediators derived from PUFAs such as AA, EPA, and DHA, and recent studies have described the involvement of eicosanoids and SPMs with versatile functions in skin wound healing (Table 1). For example, among AA metabolites, several COX or CYP products demonstrated a positive effect on skin wound healing through various activities, including modulation of clotting, macrophage polarization, keratinocyte migration, and angiogenesis, and the metabolites produced by the 5-LOX pathway such as LTs delayed the process of skin wound healing partially via the mechanisms of excessive inflammation, including neutrophil recruitment and reduced macrophage polarization to M2 phenotype. Furthermore, SPMs accelerated wound healing, as indicated by the promotion of resolution, without causing defects in the initial inflammation. As it is relatively easy to administer topical treatments to skin wounds, targeting the eicosanoid and SPM signals could be promising as an alternative strategy for patients with acute injury or surgery-induced injury as well as chronic wounds with aberrant inflammation or impaired vascularization.

However, in general, eicosanoids are rapidly degraded by non-enzymic pathway or enzymatic metabolism in vivo [134,135]. Thus, the development of a formulation strategy that could prolong the release of eicosanoids might have a better effect on tissue repair and regeneration [136]. Indeed, the topical application of PGE_2_ containing chitosan hydrogel or microparticle-encapsulated LXA_4_ demonstrated improved efficacy compared to PGE_2_ or LXA_4_ alone [77,125].

On the other hand, wound healing might be promoted by the inhibition of eicosanoids production such as LTs that have a negative effect on the processes [110,111,112]. An aspect that must be noted is that because eicosanoids and SPMs have a common precursor, modifying one pathway will affect the other. Therefore, a much more comprehensive study such as a time-course study of lipidomics during skin wound healing is required to better understand the complex interaction of lipid mediators.

Furthermore, the specific receptors of several lipid mediators, including EETs, and the majority of SPMs are still ambiguous. GPR40, also known as free fatty acid receptor-1 (FFAR-1), is a low-affinity receptor of EETs and mediates vascular actions with micromolar concentration of EETs [137]. Regarding SPM receptors, it has been reported that LXA_4_, RvD1, and RvD3 can stimulate both ALX/FPR2 and DRV1/GPR32 receptors [138]. In addition, recent studies have demonstrated that GPR37 and LGR6 are the receptors for PD1 and MaR1, respectively [139,140]. However, the receptors for the other subtypes of SPMs have not been identified, and it is also not clear whether EETs or SPM/receptor signals are functional under physiological or pathological conditions. To gain a better understanding of the physiological functions of these mediators and receptors for the development of clinical applications, it is indispensable to clarify the molecular mechanisms underlying the phenomenon of wound healing.

## Figures and Tables

**Figure 1 ijms-21-08435-f001:**
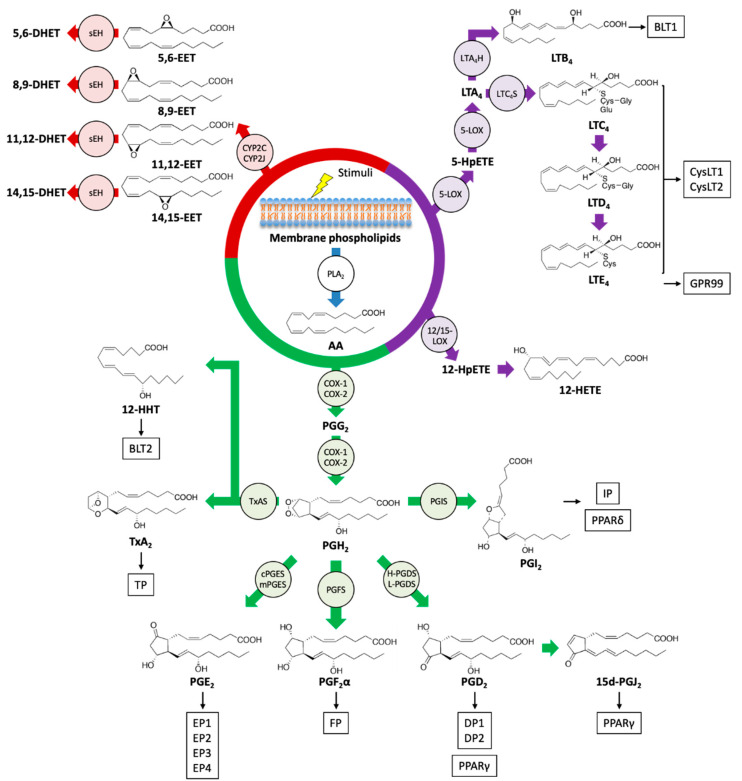
Biosynthetic pathways and receptors of eicosanoids. Upon stimulation, an omega-6 PUFA arachidonic acid (AA) is liberated from the phospholipids of the cell membrane by the action of phospholipase A_2_ (PLA_2_), and AA is metabolized into various bioactive lipid mediators. Prostanoids, leukotrienes (LTs), hydroxyeicosatetraenoic acid (HETEs), and epoxyeicosatrienoic acid (EETs) are formed from AA via cyclooxygenase (COX) (green), a lipoxygenase (LOX) (purple), and cytochrome P450 (CYP) (red) pathways, respectively. Specific eicosanoid receptors and peroxisome proliferator-activated receptors (PPARs) that are potentially activated by eicosanoids are shown in boxes. sEH: soluble epoxide hydrolase; TxAS: thromboxane A synthase; PGIS: prostacyclin synthase; PGFS: prostaglandin F synthase; cPGES: cytosolic prostaglandin E synthase; mPGES: membrane-associated prostaglandin E synthase; H-PGDS: hematopoietic-type prostaglandin D synthase; L-PGDS: lipocalin-type prostaglandin D synthase; LTA_4_H: LTA_4_ hydrolase; LTC_4_S: LTC_4_ synthase; HpETE: hydroperoxyeicosatetraenoic acid.

**Figure 2 ijms-21-08435-f002:**
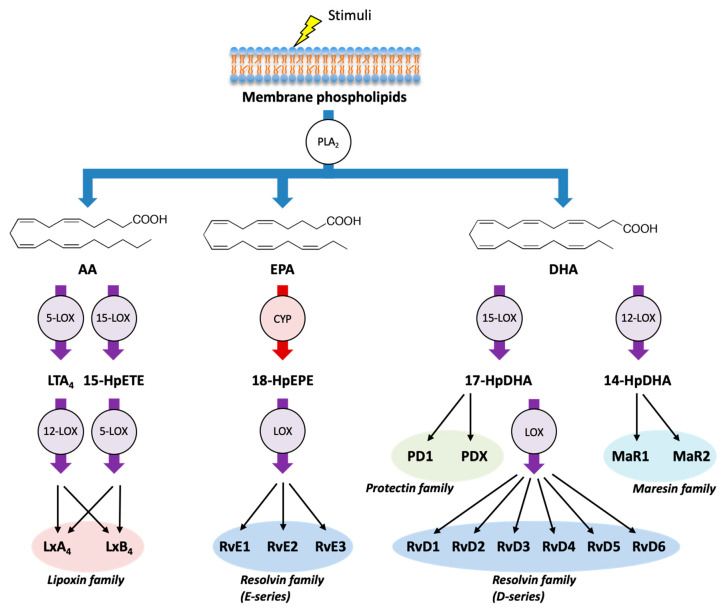
Biosynthetic pathways of specialized pro-resolving mediators (SPMs).In addition to AA, LOX pathway (purple) and CYP pathway (red) can metabolize the omega-3 PUFAs eicosapentaenoic acid (EPA) and docosahexaenoic acid (DHA) into various SPMs, which act as endogenous immunoresolvents upon stimulation. Lipoxins (LxA_4_ and LxB_4_) are formed from AA, E-series of resolvins (RvE1, RvE2, and RvE3) are formed from EPA, and D-series of resolvins (RvD1, RvD2, RvD3, RvD4, RvD5, and RvD6), protectins (PD1 and PDX), and maresins (MaR1 and MaR2) are formed from DHA. HpEPE: hydroperoxyeicosapentaenoic acid; HpDHA: hydroperoxydocosahexaenoic acid.

**Figure 3 ijms-21-08435-f003:**
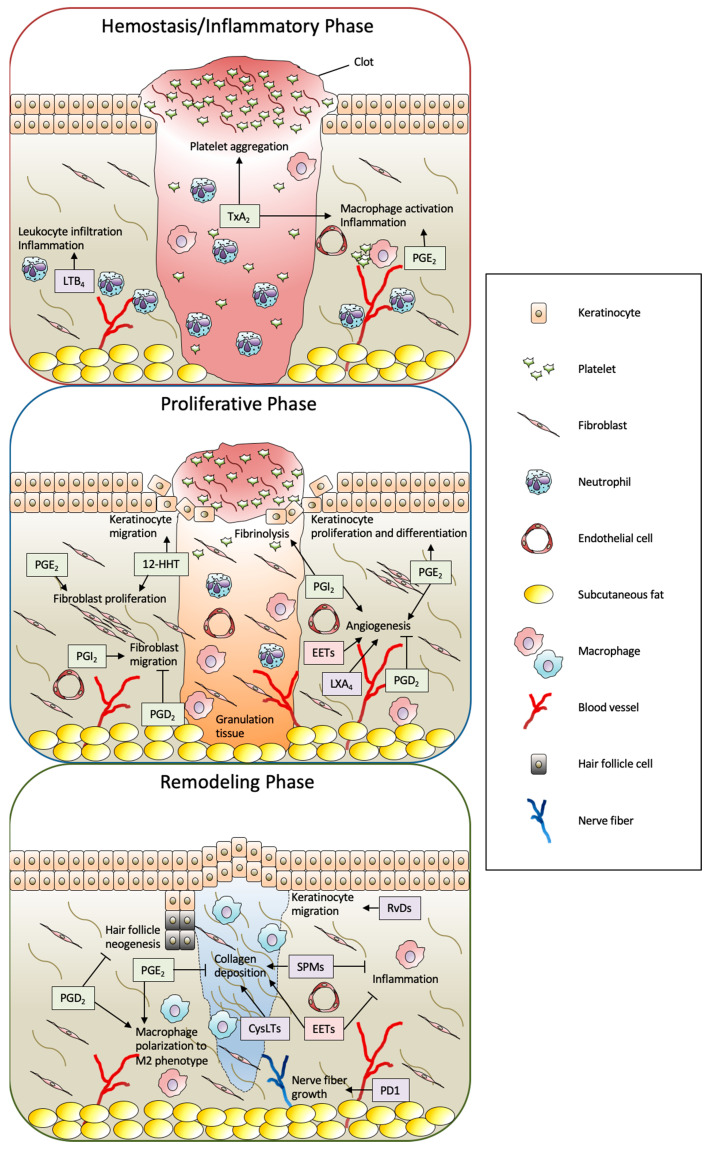
Summary of biological functions of eicosanoids and SPMs in full-thickness wound healing. The process of skin wound healing consists of three consecutive and overlapping steps, including hemostasis/inflammatory, proliferative, and remodeling phases. In this process, a wide variety of eicosanoids and SPMs are produced from skin-resident cells and infiltrated cells by the wound stimuli and play multiple roles. The metabolites formed by the COX, LOX, and CYP pathways are shown as green, purple, and red boxes, respectively. PG: prostaglandin; Tx: thromboxane; LT; leukotriene; 12-HHT: 12(*S*)-hydroxyheptadeca-5*Z*,8*E*,10*E*-trienoic acid; EET: epoxyeicosatrienoic acid; LX: lipoxin; Rv: resolvin; PD1: protectin D1; SPM: specialized pro-resolving mediator; CysLT: cysteinyl leukotriene.

**Table 1 ijms-21-08435-t001:** Overview of eicosanoids and SPMs involved in skin wound healing.

Eicosanoid and SPM	Source	Biosynthetic Pathway	Function Related to Skin Wound Healing	Reference
TxA_2_	AA	COX-1/2 TxAS	Platelet aggregation (Hemostasis) ↑ Inflammation ↑ Endothelial migration and angiogenesis ↑ (in vitro)	[50,51] [54] [55,56]
12-HHT	AA	COX-1/2 TxAS	Keratinocyte migration ↑ Fibroblast proliferation ↑	[58] [59]
PGE_2_	AA	COX-1/2 cPGES or mPGES	Angiogenesis ↑ Fibroblast proliferation ↑ Inflammation ↓ Keratinocyte proliferation and differentiation ↑ Macrophage polarization to M2 phenotype ↑ Collagen synthesis/Fibrosis ↓	[65,66] [65,66] [69] [70,71,72,73,74] [76,77] [78]
PGD_2_	AA	COX-1/2 H-PGDS or L-PGDS	Cutaneous barrier function ↑ Macrophage polarization to M2 phenotype ↑ Fibroblast migration ↓ (in vitro) Angiogenesis ↓ Hair follicle neogenesis ↓	[82,83] [84,85,86,87,88] [89] [90] [91,92]
PGF_2_α	AA	COX-1/2 PGFS	Unknown	-
PGI_2_	AA	COX-1/2 PGIS	Fibrinolysis ↑ Fibroblast migration ↑ (in vitro) Angiogenesis ↑	[101] [101] [102,103]
LTB_4_	AA	5-LOX LTA_4_H	ROS production ↑ Inflammation ↑ Macrophage polarization to M2 phenotype ↓	[110] [111] [112]
CysLTs (LTC_4_, LTD_4_, and LTE_4_)	AA	5-LOX LTC_4_S	ROS production ↑ Inflammation ↑ Macrophage polarization to M2 phenotype ↓ Collagen deposition ↑	[110] [111] [112] [113]
12-HETE	AA	12/15-LOX	Keratinocyte proliferation and migration ↑ (in vitro)	[115,116]
EETs (5,6-, 8,9-, 11,12-, and 14,15-EET)	AA	CYP2C or CYP2J	Angiogenesis ↑ Inflammation ↓ Collagen deposition ↑	[118,121,122] [119,120,121] [121]
LXA_4_	AA	5-LOX 12- or 15-LOX	Inflammation ↓ Angiogenesis ↑ Collagen deposition ↑ Fibroblast proliferation and migration ↓ (in vitro)	[125] [125] [125] [126]
RvE1	EPA	CYP LOX	Inflammation ↓ Collagen deposition ↑	[127,128] [129]
RvD1, RvD2, and RvD4	DHA	LOX	Collagen deposition ↑ Keratinocyte migration ↑ Fibroblast proliferation and migration ↓ (in vitro)	[129] [27] [126]
PD1	DHA	LOX	Inflammation ↓ Nerve fiber growth ↑ Re-epithelialization ↑ Collagen deposition ↑	[130] [130] [130] [130]
MaR1	DHA	LOX	Inflammation ↓ Re-epithelialization ↑ Macrophage polarization to M2 phenotype ↑	[131,132] [132] [132]

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
