# Peer review of "Eicosanoids in Skin Wound Healing"

_ijms, 2020, doi:10.3390/ijms21228435_

Round 1

Reviewer 1 Report

The ms covers an interesting topic, but there are some parts that appear to have been inserted just to fill the gap (for example too many details of the biosynthetic pathways), while I would like to read a more articulated final discussion focused on possibile applications of eicosanoids as well as on the possible modulations of eicosanoids biology in the skin, in normal and pathological conditions.

Author Response

Reviewer #1

The ms covers an interesting topic, but there are some parts that appear to have been inserted just to fill the gap (for example too many details of the biosynthetic pathways), while I would like to read a more articulated final discussion focused on possibile applications of eicosanoids as well as on the possible modulations of eicosanoids biology in the skin, in normal and pathological conditions.

[Response]

We thank the reviewer for valuable comments. We also fully agree with your opinion that it is important to understand the possible application and modulation of the eicosanoid pathway for the treatment of skin wound healing. We have added sentences to the “Summary and Perspectives” part as below. Please see line 381-387.

“However, in general, eicosanoids are rapidly degraded by non-enzymic pathway or enzymatic metabolism in vivo[134,135]. Thus, the development of a formulation strategy that could prolong the release of eicosanoids might have a better effect on tissue repair and regeneration [136]. Indeed, the topical application of PGE2 containing chitosan hydrogel or microparticle-encapsulated LXA4 demonstrated improved efficacy compared to PGE2 or LXA4 alone [77,125].

On the other hand, wound healing might be promoted by the inhibition of eicosanoids production such as LTs that have a negative effect against the processes [110-112].”

Reviewer 2 Report

The authors report an interesting review study about the role of eicosanoids in the wound healing process.

The manuscript is well written and i recommend its publication in this journal. The only suggestion is to include a short paragrph about the methodoldy used in selecting the papers and database used for bibliographic search.

Author Response

The authors report an interesting review study about the role of eicosanoids in the wound healing process.

The manuscript is well written and I recommend its publication in this journal. The only suggestion is to include a short paragraph about the methodolzy used in selecting the papers and database used for bibliographic search.

[Response]

We thank the reviewer for positive comments on the manuscript. We also appreciate your suggestion, and we have added a sentence to the “Abstract” part as below. We have used PubMed as a database for bibliographic search, and the keywords are also shown as below. Please see lines 16-18.

“This review article highlights the roles of eicosanoids on skin wound healing, especially focusing on the biosynthetic pathways and biological functions, i.e., inflammation, proliferation, migration, angiogenesis, remodeling, and scarring.”

Reviewer 3 Report

1. lines 35 & 83: "... nonenzymatic conversion ..." & "... nonenzymatically [28]." This is actually referring to eicosanoid production by eukaryotic microbes, as described in reference 28. This is still enzymatic, but it is not the action of endogenous enzymes.

2. line 74: Prostacyclins should be included as bioactive prostanoids.

3. line 77: growth --> proliferation

4. lines 386-389: Add Author Contributions, Funding, Acknowledgements and Conflict of Interest.

Author Response

  1. lines 35 & 83: "... nonenzymatic conversion ..." & "... nonenzymatically [28]." This is actually referring to eicosanoid production by eukaryotic microbes, as described in reference 28. This is still enzymatic, but it is not the action of endogenous enzymes.

[Response]

We thank the reviewer for pointing this out to us. We sincerely apologize for confusing you. It is known that PGH2 could be degraded into PGD2, PGE2, PGF2α, or 12-HHT in vitro even in the absence of any enzymes. However, as you mentioned, this phenomenon has not been proven in vivo. Thus, we have deleted the words “nonenzymatic” and “nonenzymatically”. Please see lines 36 and 88.

  1. line 74: Prostacyclins should be included as bioactive prostanoids.

[Response]

We appreciate your suggestion, and this has been done as below. Please see line 79-81.

“The family of bioactive prostanoids includes prostaglandins (PGD2, PGE2, PGF2α, and PGI2) and TxA2 [21].” --> “The family of bioactive prostanoids includes prostaglandins (PGD2, PGE2, PGF2α), prostacyclin (PGI2), and TxA2 [31].”

  1. line 77: growth --> proliferation

[Response]

We appreciate your suggestion, and this has been done. Please see line 83.

  1. lines 386-389: Add Author Contributions, Funding, Acknowledgements and Conflict of Interest.

[Response]

Thank you for pointing it out, and this has been done as below. Please see lines 403-409.

Round 2

Reviewer 1 Report

I suggest to accept the ms

Author Response

We appreciate the reviewer for critical comments.

To facilitate your review of our revisions, the following is a point-by-point response to the questions and comments delivered in your letter dated November 3.

Lines 81 and 90: Please replace oxidized with oxygenated

[Response]

We agree with your suggestion and this has been done. Please see Lines 83 and 92.

Line 197: the authors refer to M1 macrophages as scavengers; however, I believe M2 macrophages are associated with scavenging activity. Could the authors substantiate their claim?

[Response]

We used “scavenger phenotype” as a term to describe the phagocytic function of M1 macrophages which destroys/eliminates pathogen or damaged cells. However, as you mentioned, M2 macrophages also have a scavenging activity, and especially scavenger receptor for the hemoglobin-haptoglobin complex, CD163, is widely used as an M2 macrophage marker. Thus, to avoid ambiguity, we revised the sentence as follows.

Line 205: “the M1 scavenger phenotype macrophage” --> “the pro-inflammatory M1 phenotype macrophage”

Lines 271-272: The authors state that “LTE4 was found to enhance vascular permeability through the activation of GPR99 in the skin” I do not think that any of the cited references show that GPR99 is expressed in skin. The authors should either revise or provide references that show GPR99 expression in healthy or wounded skin and whether GPR99 mediates vascular permeability via LTE4 in skin.

[Response]

We agree with your comment. As you mentioned, Kanaoka et al. (2013) [Reference number 109] did not identify the GPR99 expressing cell in the skin. However, the authors demonstrated that the intradermal injection of LTE4 into the ear induced vascular permeability through GPR99. Thus, we revised the sentence as follows.

Lines 280-281: “LTE4 was found to enhance vascular permeability through the activation of GPR99 in the skin [108,109].” -->       “the intradermal injection of LTE4 induced vascular permeability in WT-mice ear but not in GPR99-dificient mice ear [108,109].”